# Exploring Cyberaggression and Mental Health Consequences among Adults: An Italian Nationwide Cross-Sectional Study

**DOI:** 10.3390/ijerph20043224

**Published:** 2023-02-12

**Authors:** Giuseppina Lo Moro, Giacomo Scaioli, Manuela Martella, Alessio Pagani, Gianluca Colli, Fabrizio Bert, Roberta Siliquini

**Affiliations:** 1Department of Public Health Sciences and Pediatrics, University of Turin, 10126 Turin, Italy; 2AOU City of Health and Science of Turin, 10126 Turin, Italy

**Keywords:** cyberaggression, cyberviolence, cyberbullying, mental health, adults

## Abstract

Cyberaggression (CyA) embraces a broad spectrum of hostile behaviors through electronic means. This cross-sectional study aimed to evaluate features and outcomes of this phenomenon among Italian adults. A nationwide survey was distributed on social media platforms. Being victim and being perpetrator of CyA were the primary outcomes; positive scores for GAD-2 (generalized anxiety disorder) and PHQ-2 (depressive symptoms) scales were the secondary outcomes. In total, 446 surveys were collected. Considering the primary outcomes, 46.3% and 13.5% reported having been victims and perpetrators of CyA, respectively. Politics, ethnic minority, and sexual orientation were main subjects triggering CyA. A higher likelihood of being cyber-victims was observed for women and the LGBTQA+ group. Women were less likely to be CyA perpetrators. There was an association between being a CyA victim and a CyA perpetrator. A total of 22.4% and 34.0% respondents scored positive for PHQ-2 and GAD-2, respectively. The main mental health consequences after CyA exposure were anger and sadness, whereas sleep alterations and stomach ache were the most experienced psychosomatics symptoms. No significant relationships between PHQ-2/GAD-2 and CyA emerged. CyA also represents a crucial public health issue among Italian adults. Further investigations are needed to better define the phenomenon and to study the potential consequences on mental health.

## 1. Introduction

The widespread availability of Internet connections and online applications allows people from all around to be in touch and communicate with each other in real time. In the digital era, personal behaviors and social relationships have moved to the online context, leading to a constant increase in human interactions in cyberspace. In addition to sharing information and opinions, controversial behaviors have spread among internet users [1]. Therefore, violence and other forms of aggression frequently recur on online social platforms, and similarities between traditional face-to-face (FtF) and online assault are often observed [2].

Since the mid-1980s, the Council of Europe began to work on crimes committed via the Internet and other computer networks, gradually negotiating an international treaty, opened for signature in 2001. The agreement, known as the “Budapest Convention on Cybercrime”, remains the most relevant international document on cybercrime; it provides for the criminalization of offences entailing electronic evidence and for appropriate legal tools, fostering countries to pursue a common criminal policy [3]. However, in Italy, a specific law to address cyber-aggressive behaviors targets only minors and cyberbullying, leaving adults and minority groups substantially with scarce safeguard against online aggressions [4]. According to the Cybercrime Convention Committee’s (T-CY) mapping study, defining cyberviolence (CyV) is an ongoing challenge due to its diverse and multiple ways of expression. The study aimed to identify and describe acts of CyV, to supply experience and strategies adopted by state parties, and finally, to develop recommendations and further actions. CyV is defined as committing violence through computer systems, consequently damaging people in various ways (physically, sexually, psychologically, or economically), by exploiting the characteristics or vulnerabilities of individuals. Hence, acts identifiable as CyV involve a variety of offences and harassment, whose definitions overlap or intersect each other. For instance, T-CY included cyberbullying under the cyber-harassment subcategory, eventually suggesting CyV as a comprehensive term to describe aggressive acts in cyberspace [5].

Accordingly, the spread of CyV episodes and the consequences affecting the wellbeing of people involved have encouraged researchers to consider CyV a serious public health issue, worthy of further investigations [6,7].

Overall, since FtF bullying has been recognized as a threat for youth mental health [8], research on CyV has focused on children and adolescents. The youngest generations have been increasingly exposed to new communication technologies and, over the years, bullying has translated into cyberspace; previous findings suggested that internet harassment spreads significantly together with the daily and popular use of social media and digital technologies [9]. Moreover, cybervictimization was shown to have an impact on mental health during development [10]. As Bottino et al. observed, cyberbullying affects 10–20% of adolescents, entailing emotional stress and mental disorder [11]. Analyses across several age groups highlighted that CyA episodes entail different dynamics between adults and adolescents [12].

However, the literature about adults remains scarce, with a few studies conducted in New Zealand and Canada [6,10], and data about prevalence among older age groups are lacking. In addition, the characteristics of this phenomenon can be different across countries, because of cultural factors and social attitudes. For instance, the role of women within patriarchal systems is subjected to morality taboos and stigmas, although women are the targeted victims [13]. Thus, this paper primarily aimed to evaluate the extent and features of cyberaggression (CyA) episodes among Italian adults, trying to fill the gap existing in the scientific literature, and to better clarify the impact and epidemiology of this emerging threat. In addition, it aimed to explore variables associated with such episodes to identify potentially vulnerable groups. Indeed, although in Italy in 2017 the Italian Parliament approved a law in support of minors and their families [4] and online hate speech against groups of population (i.e., women, refugees, and sexual and gender minorities), which roused nongovernmental organizations, media, and public opinion [14], studies about this issue have not yet been implemented. Secondarily, the present paper attempted to assess the relationship between CyA and negative mental health outcomes among adults, thus exploring potential similarities with the findings that the scientific literature reported concerning youth [10,11].

The final goal of this work was to raise awareness among the general population and institutions through a better outlining of CyA features and consequences among adults.

## 2. Materials and Methods

### 2.1. Study Design and Recruitment

A nationwide cross-sectional study was carried out between 13 April and 20 June 2021, through an online survey distributed on social media platforms (mainly Facebook, Twitter, and WhatsApp). All procedures were in accordance with the 1964 Helsinki declaration and its later amendments. The protocol of the study was approved by the Ethics Committee of the University of Turin.

The target population was adult people (from the age of 18 onward) currently living in Italy. Raosoft^®^ was used to determine that the minimum sample size was 385, considering a 5% margin of error, 95% confidence level, 50% response distribution, and the size of the Italian adult population. Participants were enrolled through convenience sampling. Informed consent was required to access the questionnaire. Participation was voluntary, anonymous, and without compensation.

In the present study, CyV and CyA were used as synonyms indicating every kind of act perpetrated to damage, insult, or harass people via Internet, especially through online social media [3]. Considering violence occurring in FtF situations, offline, in-person, and FtF violence were used as synonyms.

### 2.2. The Questionnaire

A survey was developed by the researchers after a preliminary literature review on existing research about forms of CyA and their potential impact on mental health among diverse age categories [13,15,16]. Questions were organized in four main sections to gather sociodemographic information, personal background, CyA and online behaviors, mental health, and other consequences of CyA. Overall, multiple-choice questions were used.

The first section aimed to define sociodemographic features, investigating data about age, gender, sexual orientation (the options allowed categorization into heterosexual and lesbian, gay, bisexual, transgender, queer, asexual, and other, e.g., pansexual: LGBTQA+), education, employment, and nationality. Indeed, demographic features can catalyze CyV, and hate materials typically aim todespise and teach collective because of gender, sexuality, and ethnicity [17,18].

The second section focused on personal background concerning family and social relationship currently and during developmental ages (childhood and adolescence), socioeconomic status, risk of alcohol and drug abuse, and frequency of exposure to social networks. Overall, these items were chosen because, in previous research among teenagers, parenting and family relationships have been associated with bullying and cyberbullying. Poor emotional relationship and lack of affection and emotional communication could be considered predictors of violent behaviors and victimization [19]. In addition, recent findings suggested that drug and alcohol abuse represents a risk factor of perpetrating CyA [20]. Alcohol and substance consumption among adolescents may predict an increased risk of peer victimization (both FtF and online), although a bidirectional relationship is still uncertain. Alcohol drinking and binge drinking showed a correlation with engaging in online aggressive behaviors, as both predictors and consequences of CyA [21]. To explore an association between this issue and CyA, the CAGE-AID validated scale (Cut Down, Annoyed, Guilty, and Eye Opener—Adapted to Include Drug Use) was included in the survey. It enabled screening the risk of both alcohol and drug abuse; by answering “yes” at least to one question, further medical evaluation for diagnosis is recommended [22,23].

The third part aimed to define whether interviewees experienced, at least once in their life, CyA episodes as a bystander, perpetrator, and/or victim. A list of multiple CyA acts allowed one or more answers among harassment, defamation, sharing of intimate content without consent, stalking, discrimination and hate speech, insult, identity theft, and other. Since many platforms have been often the scene of perpetrating CyV [7], this section explored where and how respondents experienced such episodes (social networks and means such as private message, audio message, image, meme, post, comment, video, email, or other). Recurring themes (politics, science, ethnic minorities, women, sexual orientation, gender identity, disability, religion, physical appearance, celebrities, and others) were also inquired to better understand and depict this phenomenon. Furthermore, since episodes of cyberbullying often overlap with traditional bullying, considering both cyberbullies and cybervictims [11], experiences of in-person aggressions were explored.

The perception of CyA frequency before and after the COVID-19 pandemic was considered as other studies found an effect of the pandemic on cyberbullying behavior [24,25].

In the fourth section, since the evaluation of the psychophysical wellbeing of participants represented a crucial approach in previous research about CyA [13,15], mental health symptoms were assessed through validated scales (Patient Health Questionnaire-2: PHQ-2 and Generalized Anxiety Disorders-2: GAD-2), along with a list of questions about emotional status and psychosomatic symptoms after CyA exposure. The PHQ-2 is a two-item instrument for depression screening. A score of three or above relates to a higher probability of depression and is identified as the cutoff point for additional evaluation [26]. Screening for anxiety disorders was performed through the GAD-2, a two-item questionnaire. A score of three or above relates to a higher probability of an anxiety disorder and is an indication for further evaluation [27]. The Italian official version of both scales was adopted [28].

For a detailed description of the potential impact of CyA, the research team elaborated a list of questions about symptoms, feelings, and emotional status after CyA episodes rated through Likert-like scale answers (one possible option from not at all, a little, moderately, and a lot). A 13-item query attempted to assess the perceived presence of fatigue, anhedonia, sadness, anxiety, rumination, anger, fear, altered eating patterns, sense of guilt, suicidal thought or attempt, revenge, pride, and sense of belonging. Then, a six-item query evaluated psychosomatic symptoms such as muscle tension and rigidity, headache, stomach ache, gastrointestinal symptoms as nausea and vomit, immunodeficiency signs, and sleep alterations.

Additionally, respondents were asked about experiencing further troubles after CyA exposure (damaging of own reputation, sentimental and familiar contacts, workplace issues, reducing or renouncing usage of social network, change in private life, and legal proceedings and expenses). Lastly, information about use of mental health services before and/or after CyV episodes (independently from having experienced cyberaggression as victim, bystander, or perpetrator) and CyV-related legal advice was gathered.

### 2.3. Statistical Analysis

Descriptive analysis was performed for all variables. Since the Shapiro–Wilk test reported non-normal distributions, the median and interquartile range (IQR) were used to describe continuous variables.

Being victims and being perpetrators of CyA were the primary outcomes. Scoring positive on the GAD-2 scale (as increased risk of general anxiety disorder) and scoring positive on the PHQ-2 scale (as increased risk of major depressive disorder) were the secondary outcomes.

All outcomes were binary variables. Chi-square tests and Mann–Whitney tests were performed to determine differences between the groups defined by each outcome.

Multivariable logistic regressions (adjusted for age and gender) were elaborated to explore the effects of variables on the binary outcomes. In the multivariable model, the following independent variables were entered: sexual orientation, education level, job, place of birth (considering also place of birth of parents), relationship with family during childhood, social relationships in adulthood, economic situation, alcohol or drug abuse, in-person bullying (experienced as victim or perpetrator), use of mental health services, being a victim of CyA, and being a perpetrator of CyA. Results were expressed as the adjusted odds ratio (adjOR) and 95% confidence interval (CI).

For all analyses, the software SPSS (Statistical Package for Social Science, version 27) was used. Missing values were excluded. Considering results as statistically significant required *p* < 0.05.

## 3. Results

### 3.1. Characteristics of the Sample

A total of 446 surveys were collected. The median age was 32 years (IQR = 26–44). Interviewees were mostly women (61.7%). A total of 12.6% belonged to the LGBTQA+ group. Only 4.7% of participants were born in a foreign country or had foreign parents (one or both). Almost 27% scored positive on CAGE-AID scale, and a conflicting family relationship was reported by almost 26%. The majority stated the intensive use of both social networks and messaging applications (72.6% and 92.6%, respectively), whereas other platforms were barely used. Table 1 and Table 2 show a detailed description of the sample.

Considering the primary outcomes, 46.2% stated they had been a victim of CyA and 13.5% stated having been a perpetrator of CyA. A total of 33.4% were victims only and a total of 1.4% were perpetrators only (12.1% were both). Only 22.9% had never witnessed CyA as a bystander. Regarding the overlap with bystanders, 95.1% of victims and 96.7% of perpetrators were also bystanders.

Among victims, the most frequently reported CyA was insult (27.4%) and harassment (20.2%). According to perpetrators and bystanders, insult was also the most experienced CyA act (11.1% and 63.9%, respectively). Moreover, bystanders witnessed discrimination (63.9%) and defamation (48.4%).

Social networks were almost unanimously (96% of respondents) the foremost platform where CyA acts occurred (among these, 71.6% on Facebook). According to participants’ opinion, CyV was mainly expressed through comments and posts (83% and 67.2%, respectively), and 41.7% of them found that CyV episodes increased during the pandemic. Politics, ethnic minority, female gender, and sexual orientation were the main popular subjects triggering CyA according to more than 50% of respondents (Figure 1 and Appendix A).

Regarding the secondary outcomes, PHQ-2 and GAD-2 scores suggested a need for further medical evaluation for 22.4% and 34% of the sample, respectively.

About 42% of respondents endured FtF violence during their childhood; among them, 15.8% also engaged in perpetrating CyA, whereas 51.9% were cyber-victims as well. During adulthood, the percentage of a FtF victim was considerably lower (4.3%); nevertheless, 57.9% of them were also victims of CyV. According to 52% of interviewees, violent behaviors occurred more frequently online than in person (Appendix A).

Regarding the impact of CyA on mental health after CyA episodes, sadness and anger were reported by 60% of participants, whereas almost half of them declared having experienced anxiety and rumination. Pride and revenge feelings were admitted by one-fourth of respondents, and sleep alterations and stomach ache were the most experienced psychosomatic symptoms. Moreover, the investigation about their private lives revealed that the most frequent consequences after CyA were relationship conflicts for both cyber-perpetrators and cyber-victims (Appendix A).

### 3.2. Relationships between the Outcomes and Participants’ Characteristics

Cyber-victims and cyber-perpetrators were allegedly younger than people never involved in such episodes. However, age was not differently distributed across victims and perpetrators (*p* = 0.176). LGBTQA+ individuals showed a higher risk of being cyber-victims and cyber-perpetrators compared to the heterosexual group (Table 1 and Table 2).

CyV perpetrators reciprocally showed a higher probability of being victims, and vice versa, especially when committing acts as harassment, discrimination, and insult.

Comparing online and offline contexts, experiencing FtF violence during both child- and adulthood was positively associated with being victim of CyA. Indeed, victims of FtF violence were also victims of CyA, in comparison to those who never experienced FtF aggression. Cyber-aggressors were more likely to experience FtF violence as victims, bystanders, and perpetrators (Appendix A).

Chi-square tests showed positive associations between cybervictimization and mental health-related symptoms such as fatigue, abulia and anhedonia, anxiety, anger and rumination, sense of guilt, and suicidal thoughts or attempts. Accordingly, all inquired psychosomatic symptoms were also positively associated with being victims of CyA. Committing cyberviolence was positively associated with some mental health and psychosomatic symptoms such as rumination, anger, and headache.

Participants who scored positive on both PHQ-2 and GAD-2 were younger than those who scored negative. A higher education level and being students were significantly associated with a higher risk of generalized anxiety disorders and depressive symptoms. Positive scores on GAD-2 and PHQ-2 resulted associated with conflictual family relationship and poor social contacts, as well as with risk of alcohol and substance abuse.

Cyber-victims were positively associated with suffering from anxious symptoms after exposure to harassment, defamation, nonconsensual sharing of intimate content, discrimination, and insult. Discrimination and hate speech were also positively associated with a positive PHQ-2 score. Participants who witnessed nonconsensual sharing of intimate content and stalking were more likely to score positively on GAD-2 and PHQ-2 scales (Appendix A).

### 3.3. Multivariable Models

Concerning primary outcomes, a positive association between online and offline violence was found, for both victims and perpetrators. Indeed, committing FtF violence was associated with being a CyA perpetrator, and being a victim of FtF aggression was associated with being a CyA victim. CyA perpetrators were also more probably victims of CyA (and vice versa). Females had a lower probability of being CyA perpetrator and a higher likelihood of being a CyA victim. Belonging to the LGBTQA+ group and being positive for CAGE-AID showed an increased likelihood of reporting to be CyA victims. Lastly, having access to mental health services was positively associated withbeing a CyA perpetrator.

Regarding secondary outcomes, multivariable analyses highlighted a higher risk of anxious symptoms for female gender, students, and those who accessed any mental health service. Furthermore, social isolation and loneliness were predictors of anxiety and depressive disorders. Furthermore, a positive CAGE-AID score was significantly related to both secondary outcomes (Table 3 and Table 4).

## 4. Discussion

The current study primarily aimed to define the extent and features of CyA among adult population living in Italy, exploring potential associations between CyA and sociodemographic factors, personal experiences, and traits. Secondarily, our research tried to outline the impact on adults’ mental health of this recent expression of violence.

Our findings revealed that 46.3% of respondents were subjected to CyA and 77.1% witnessed such episodes at least once in their life, with even 13.5% of them confessing to having been a perpetrator. Furthermore, people involved in CyA episodes were younger than people who did not, consistent with previous research [5,29,30]. A recent survey conducted among adults in New Zealand reported that 14.9% of participants had been the target of cyberbullying at least once in their lifetime. More than 40% of respondents belonging to the 18–25 age cohort reported having experienced cyberbullying, whereas prevalence gradually decreased across older age cohorts [6]. As suggested by Feinstein et al., data from the literature about the prevalence of cybervictimization ranged from 4% to 39% of youth and 9% to 43% of young adults, with wide differences across age, sampling, timeframe, and measure [29]. A cross-national six-country study analyzed rates of 10–20% of young adults admitting to having previously produced online hate content [30].

As explained in Section 2, we explored gender, sexual orientation, and ethnicity, as these features may be potential targets of CyA [17,18]. First, the present study showed a higher risk of being victim of CyA for the LGBTQA+ group, in line with other findings. According to the Anti-Defamation League (ADL) report “Online Hate and Harassment—The America Experience 2021” (an annual nationally representative survey), a higher rate of overall harassment (64%) was reported by LGBTQ+ respondents compared with all other identity groups in the USA [31]. Moreover, in our analyses, female gender also appeared more frequently targeted by CyA. Several studies drew contrasting conclusions about gender difference in online victimization and violence perpetration, although cyberdating violence affects mostly women. Nonetheless, similar results in higher victimization rates among females were reported [32,33]; conversely, findings about a wider targeting of males in online aggressions was also described [6,7,13,18].

Concerning ethnicity, no significant results emerged, potentially because of the small size and poor representativeness of the foreign subgroup in our sample. Indeed, a small group of participants (4.7%) were born in a foreign country or from foreign parents, whereas 8.4% of the Italian population came from abroad [34]. Moreover, it is possible that, in the Italian context, which has been poorly explored before the present study, the CyA phenomenon has specific features due to the sociocultural characteristics of the country. Accordingly, a systematic review about the impact on adult mental health of cyber-abuse highlighted, in most studies, conflicting findings about the prevalence of victimization rates concerning gender and ethnicity due to inadequate representativeness of the samples [7]. Furthermore, a large study involving six countries highlighted different major foci of CyV across the countries, thus suggesting that CyV traits may vary in different contexts [35]. For instance, online hate addressed mainly sex or gender identity among Spanish respondents, whereas ethnic minorities were a major focus of online violent speech for participants from the UK, the USA, and France [35]. It is worth noting that, in addition to ethnicity, the participants of our study considered sexual orientation, female gender, and political opinions as the major triggers of CyV. Similarly, other data showed that online hate comments targeted mainly immigrants, women, sexual and gender minorities, and politicians and political opinions [16].

Consistent with other research about a possible overlap among online and traditional bullying [11], in the present study, a significant association between online and offline victimization was found, along with a higher tendency of CyV among offline aggressors. Indeed, it should be highlighted that the vulnerable groups that we described in the previous paragraph were reported as also being vulnerable groups for offline hate. The Italian Institute of Statistics (ISTAT) reported that 31.5% of women aged 16–70 stated having been victims of physical or sexual violence, and 16.1% stated having experienced psychological violence and stalking [36]. Moreover, a committee established by Italian Parliament reported that 25% of Italians reckoned homosexuality a disease and 65% considered refugees a social and economic burden [37]. Lingiardi et.al (2020) suggested that online hate speech against certain minorities mirrored daily social and political debates and events [38].

As for the main features of CyA, interviewees highlighted social networks as the platform where CyA episodes happen more frequently (especially Facebook), in line with previous international results [35,39]. Indeed, a survey carried out by the Pew Research Center in 2014 found social media sites to be the most common location where people experienced online harassment, especially young adults aged 18–29 years and women. Furthermore, most of participants of an international cross-sectional study identified hate materials on Facebook [35,39]. These findings reinforce the need of studying strategies to tackle CyA in these contexts. In addition, our sample declared that CyA showed up more frequently through insults, discrimination, defamation, and harassment. Victims, perpetrators, and bystanders identified insult as the most frequent act. In this regard, DeMarsico et al. assessed motivations of adult CyA: most endorsed types of CyA were flaming (“defined as posting or sending offensive materials, e.g., hostile or insulting messages”) and trolling (“defined as behaving in a deceptive, destructive, or disruptive manner”), both reported by over 50% of the sample [2].

Interestingly, our findings highlighted an increased risk of alcohol and substance abuse among CyA victims and perpetrators (although it was confirmed in the multivariable analysis only among victims). A recent meta-analysis suggested alcohol as the strongest risk factor for online violence acting, compared to other substances [20]. So far, a deep assessment of online aggressive behaviors, as well as victimization related to increasing impulsivity and risk of addiction, is lacking, and further evaluations of this issue are needed.

Lastly, considering our secondary outcomes, the multivariable analyses did not reveal a higher probability of being victims and/or perpetrators of CyA among people who scored positive on GAD-2/PHQ-2. Their limited time span of investigation (i.e., the previous 2 weeks) could explain these findings. Interestingly, the chi-squared analyses found significant associations between cybervictimization and emotional distress (anxiety, anger, abulia, anhedonia), fatigue, and psychosomatic symptoms. A recent systematic review reported harmful consequences on mental health among CyA victims, along with negative effects in the private, professional, and economic sphere. It should be noted that, since most studies were online surveys, samples were not representative of victims who distrusted technology after CyA [7]. Moreover, a Canadian study showed that associations between cyberbullying victimization and adverse mental health outcomes decreased from young adulthood onward [10]. In the current study, multivariable regression models revealed a significant association with access to mental health services only for perpetrators. It is reasonable to partially explain this result considering a condition of mental distress as a predictor of dysfunctional behaviors and troubled interpersonal relationships. Previous findings showed that cyber-victims are often also cyberbullies and cybervictimization assumes cyberaggression among adolescents [9,40]. According to the strain theory, victimization induces feelings to avenge oneself through anger and frustration [13]. However, the relationship between CyA and mental health among adults should be further explored, inquiring into the reciprocal role of victims and perpetrators.

The present study had some strengths and limitations. To our knowledge, this was the first study focused on CyV among Italian adults and its consequences on mental health. We outlined a description of this phenomenon focusing on frequency, topics, and characteristics from multiple points of view, i.e., as victims, perpetrators, and bystanders, as well as by profiling several figures of cyber-violent acts. Thus, our research provides a relevant contribute to the Italian context and encourages the consideration of CyA as a critical issue among adults. As discussed above, common traits were actually found among several previous studies, highlighting similar patterns of behaviors and attitudes across countries.

The cross-sectional design, the opportunistic sampling, and the small sample size represented the main limitations. Moreover, the questionnaire was mostly composed of items not validated through previous studies. The interpretation of our results should also consider that the sample was not fully representative of the Italian adult population for age, gender, and the percentage of foreign residents [34]. Additionally, the online survey could not categorize those who deleted their virtual identity, likely after CyV. Lastly, although validated scales were used, GAD-2 and PHQ-2 only investigate a limited timespan.

## 5. Conclusions

The present research highlighted the relevance of CyA among adults, especially in certain subgroups of the population such as women and the LGBTQA+ population, as well as a substantial overlap with in-person violence. However, further investigations are needed to better define the phenomenon and to study the potential consequences on mental health, especially in Italy where evidence is currently poor. Therefore, future research should better define epidemiological features of this growing problem, considering indeed the threatening consequences on mental health, which embrace a broad range of symptoms. As it can be considered as a public health issue, actions to increase consciousness are crucial, at both an institutional and a population level. National institutions should be involved in dissuading committing CyA, ensuring safety, and providing support for the victims. Lastly, population engagement is necessary to stem such episodes, increase awareness, and enable people to recognize episodes of online violence and hate speech.

## Figures and Tables

**Figure 1 ijerph-20-03224-f001:**
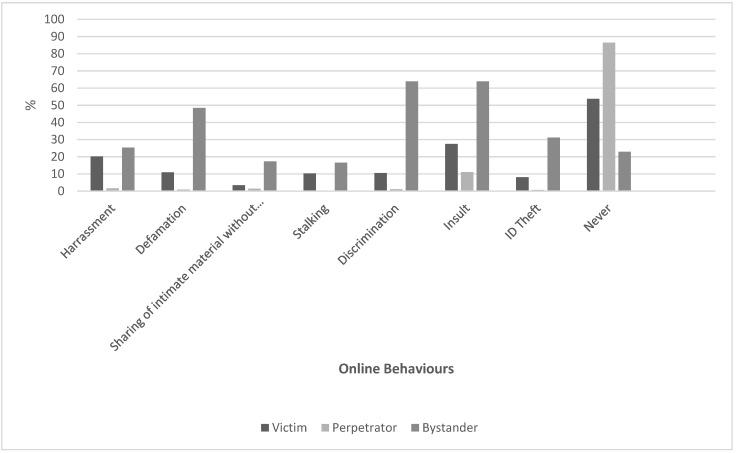
Frequency of aggressive online behaviors described stratifying by role (victim, perpetrator, and bystander).

**Table 1 ijerph-20-03224-t001:** Sociodemographic characteristics and relationships with primary outcomes.

Characteristic	Overall Sample	Being Perpetrator of CyA			Being Victim of CyA		
	N (%) (n = 446)	No N (%) 383 (85.9)	Yes N (%) 60 (13.5)	*p*-Value	No N (%) 239 (53.7)	Yes N (%) 206 (46.2)	*p*-Value
**Age**	**Median** 32	**Median** 33	**Median** 29	**0.004**	**Median** 35	**Median** 31	**0.019**
	**IQR** (26–44)	**IQR** (26–45)	**IQR** (25–36)		**IQR** (26–49)	**IQR** (25–38)	
**Gender**							
Men	167 (37.4)	133 (81.1)	31 (18.9)	**0.002**	87 (52.1)	80 (47.9)	0.192
Women	275 (61.7)	248 (90.2)	27 (9.8)		151 (55.1)	123 (44.9)	
Female-to-male ratio	1 (0.2)	1 (0.2)	0 (0)		(100)	0 (0)	
Queer	3 (0.7)	1 (33.3)	2 (66.7)		0 (0)	3 (100)	
**Sexual Orientation**							
Cis (i.e., heterosexual)	390 (87.4)	342 (88.4)	45 (11.6)	**0.002**	126 (58.1)	163 (41.9)	**<0.001**
LGBTQA+	56 (12.6)	41 (73.2)	15 (26.8)		13 (23.2)	43 (76.8)	
**Education**							
High-school diploma or lower level	248 (55.6)	216 (87.7)	30 (12.2)	0.354	132 (53.4)	115 (46.6)	**0.001**
College or Master/PhD	198 (44.4)	167 (84.8)	30 (15.2)		198 (44.5)	91 (46.0)	
**Employment**							
Employed	254 (57.0)	215 (85.3)	37 (14.7)	0.073	125 (49.2)	129 (50.8)	**0.001**
Student	145 (32.5)	122 (84.7)	22 (15.3)		77 (53.1)	68 (46.9)	
Retired, homemaker, unemployed	43 (9.6)	42 (97.7)	1 (2.3)		34 (81.0)	8 (19.0)	
**Own + parents’ birthplace**							
Italy	425 (95.3)	365 (86.5)	57 (13.5)	0.919	228 (53.8)	196 (46.2)	0.901
Born abroad or at least one parent born abroad	21 (4.7)	18 (85.7)	3 (14.3)		11 (52.4)	10 (47.6)	

*p*-Values were obtained via chi-squared test (except for age: Mann–Whitney U test). Overall sample: column percentage. Being perpetrator/victim of CyA: row percentage. Abbreviations: CyA, cyberaggression; IQR, interquartile range; LGBTQA, lesbian, gay, bisexual, transgender, queer, asexual. Significant *p*-values in bold.

**Table 2 ijerph-20-03224-t002:** Participants’ personal characteristics, online behavior, and relationships with primary outcomes.

Characteristic	Overall Sample	Being Perpetrator of CyA			Being Victim of CyA		
	N (%) (n = 446)	No N (%) 383 (85.9)	Yes N (%) 60 (13.5)	*p*-Value	No N (%) 239 (53.7)	Yes N (%) 206 (46.2)	*p*-Value
**Family relationship**							
Good/very good	331 (74.2)	293 (89.1)	36 (10.9)	**0.007**	189 (57.3)	141 (42.7)	**0.011**
Bad/very bad	115 (25.8)	90 (78.9)	24 (21.1)		50 (43.5)	65 (36.5)	
**Social life**							
Many people	126 (28.3)	103 (83.1)	21 (16.9)	0.215	63 (50)	63 (50)	0.763
Some people	293 65.7)	259 (88.7)	33 (11.3)		162 (55.5)	130 (44.5)	
No contact	16 (3.6)	12 (75)	4 (25)		8 (50)	8 (50)	
Social isolation	11 (2.5)	9 (81.8)	2 (18.2)		6 (54.5)	5 (45.5)	
**Economic status**							
High/medium	400 (89.7)	342 (86.1)	55 (13.9)	0.576	217 (54.4)	182 (45.6)	0.398
Low/very low	46 (26.5)	41 (89.1)	5 (10.9)		22 (47.8)	24 (52.2)	
**CAGE-AID**							
Negative	328 (73.5)	295 (90.5)	31 (9.5)	**<0.001**	197 (60.2)	130 (39.8)	**<0.001**
Positive	118 (26.5)	88 (75.2)	29 (24.8)		42 (35.6)	76 (64.4)	
**Social network frequency use**							
Very often/often	324 (72.6)	271 (83.6)	51 (15.7)	**0.002**	156 (48.1)	167 (51.9)	**<0.001**
Rarely/never	122 (37.4)	112 (91.8	9 (7.4)		42 (35.6)	76 (64.4)	
**Messaging apps frequency use**							
Very often/often	413 (92.6)	354 (85.7)	56 (13.5)	0.053	217 (52.5)	195 (47.2)	**0.010**
Rarely/never	33 (7.4)	29 (87.9)	4 (12.1)		22 (66.6)	11 (33.4)	
**Video platforms frequency use**							
Very often/often	178 (39.9)	143 (80.3)	33 (18.5)	**0.001**	78 (43.8)	100 (56.2)	**0.008**
Rarely/never	268 (60.1)	240 (89.5)	27 (10.1)		161 (60.1)	106 (39.9)	
**Forums/blogs frequency use**							
Very often/often	111 (24.9)	95 (85.6)	16 (14.4)	**0.034**	55 (49.5)	56 (50.5)	0.548
Rarely/never	335 (75.1)	288 (86.0)	44 (13.1)		184 (54.9	150 (44.7)	
**Dating apps frequency use**							
Very often/often	34 (7.6)	24 (70.6)	10 (29.4)	**0.002**	11 (32.3)	23 (67.6)	**0.007**
Rarely/never	412 (92.4)	359 (87.1)	50 (12.1)		228 (55.3)	183 (44.4)	
**Changes during pandemic**							
Increased	186 (41.7)	159(85.9)	26(14.1%)	**<0.001**	87 (46.8)	99 (53.2)	**0.002**
Decreased	6 (1.3)	5 (83.3)	1(16.7%)		2 (33.3)	4 (66.7)	
Unchanged	75 (16.8)	54 (72.0)	21(28.0%)		35 (46.7)	40 (53.3)	
No idea	179 (40.1)	165(93.2)	12(6.8%)		115(64.6)	63 (35.4)	
**Being victim of FtF violence**							
Never	207 (46.9)	191 (92.3)	16 (7.7)	**0.002**	138 (66.7)	69 (33.3)	**<0.001**
During childhood, adulthood, or both	234 (53.1)	191(82.3)	41 (17.7)		100 (42.9)	133 (57.1)	
**Being perpetrator of FtF violence**							
Never	385 (86.3)	349 (90.9)	35 (9.1)	**<0.001**	220 (57.3)	164 (42.7)	**<0.001**
During childhood, adulthood, or both	57 (12.8)	33 (58.9)	23(41.1)		18 (31.6)	39 (68.4)	
**Mental health service access**							
No	340 (80.8)	306 (90.3)	33 (9.7)	**<0.001**	199 (58.5)	141 (41.5)	**<0.001**
Yes	81 (19.2)	80 (19.1)	21 (26.3)		29 (36.3)	51 (63.8)	
**Risk of depression (PHQ-2)**							
Negative	333 (77.6)	294 (88.8)	37 (11.2)	0.051	184 (55.4)	148 (44.6)	0.448
Positive	96 (22.4)	78 (81.3)	18 (18.8)		49 (51.0)	47 (49.0)	
**Risk of anxiety (GAD-2)**							
Negative	283 (66.0)	251 (89.0)	31 (11.0)	0.104	165 (58.3)	118 (41.7)	0.025
Positive	146 (34.0)	121 (81.3)	24 (16.6)		68 (46.9)	77 (53.1)	

*p*-Values were obtained via chi-squared test. Overall sample: column percentage. Being perpetrator/victim of CyA: row percentage. Abbreviations: CyA, cyberaggression; FtF, face-to-face; CAGE-AID, Cut Down, Annoyed, Guilty, and Eye Opener—Adapted to Include Drug Use; GAD-2, Generalized Anxiety Disorder 2-item scale; PHQ-2, Patient Health Questionnaire 2-item scale. Significant *p*-values in bold.

**Table 3 ijerph-20-03224-t003:** Multivariable regressions models of primary outcomes.

Characteristic	Being Perpetrator of CyA		Being Victim of CyA	
	Adj OR (95% CI)	*p*-Value	Adj OR (95% CI)	*p*-Value
**Age**	0.97(0.92–1.02)	0.319	0.98 (0.95–1.01)	0.128
**Gender ^1^**				
Men	Ref.	**0.013**	Ref.	**0.048**
Women	0.37 (0.17–0.81)		1.68 (1.01–2.82)	
**Sexual orientation**				
LGBTQA+	0.50 (0.19–1.32)	0.164	4.31 (1.91–9.68)	**≤0.001**
**Education**				
High-school diploma or lower level	0.99 (0.40–2.48)	0.993	1.03 (0.57–1.89)	0.910
**Professional status**				
Employed	Ref.		Ref.	
Student	1.70 (0.57–5.11)	0.343	0.56 (0.27–1.19)	0.133
Unemployed, retired, homemaker	1.27 (0.14–3.58)	0.842	0.55 (0.19–1.61)	0.279
**Nationality**				
Born abroad or parents born abroad (1 or 2)	0.70 (0.14–3.58)	0.665	1.29 (0.44–3.78)	0.644
**Family relationship**				
Conflictual-very conflictual	1.52 (0.68–3.41)	0.306	0.89 (0.50–1.60)	0.705
**Social life**				
Many people	Ref.		Ref.	
Some people	0.51 (0.23–1.12)	0.095	1.04 (0.62–1.75)	0.889
No contact	1.11 (0.18–6.90)	0.908	0.86 (0.21–3.56)	0.834
**Social isolation**	0.96 (0.12–7.82)	0.967	0.72 (0.14–3.81)	0.701
Economic status				
Low-very low	0.60 (0.18–1.95)	0.392	1.37 (0.63–2.97)	0.421
**CAGE-AID**				
Positive	1.58 (0.75–3.30)	0.224	2.08 (1.20–3.60)	**0.009**
**Being victim of CyA**	20.39 (6.39–65.11)	**0.001**	-	-
**Being perpetrator of CyA**	-	-	15.70 (5.18–47.58)	**<0.001**
**Being victim of FtF violence ^2^**	1.04 (0.47–2.30)	0.917	1.67 (1.02–2.72)	**0.040**
**Being perpetrator of FtF violence ^2^**	3.99 (1.70–9.40)	**0.002**	0.85 (0.38–1.89)	0.697
**Access to mental health service**	2.42 (1.08–5.43)	**0.032**	1.36 (0.74–2.52)	0.321

Abbreviations: adjOR, adjusted odds ratio; CI, confidence interval; CyA, cyberaggression; FtF, face-to-face; CAGE-AID, Cut Down, Annoyed, Guilty, and Eye Opener—Adapted to Include Drug Use; LGBTQA, lesbian, gay, bisexual, transgender, queer, asexual. Notes: ^1^ Female-to-male ratio and queer were omitted due to small size. ^2^ Binary variable (never vs. during childhood, adulthood, or both). Significant *p*-values in bold.

**Table 4 ijerph-20-03224-t004:** Multivariable regressions models of secondary outcomes.

Characteristic	Risk of Depressive Symptoms		Risk of Anxiety Disorders	
	Adj OR (95% CI)	*p*-Value	Adj OR (95% CI)	*p*-Value
**Age**	0.99 (0.96–1.02)	0.538	0.99 (0.97–1.02)	0.835
**Gender ^1^**				
Men	Ref.	0.940	Ref.	**0.004**
Women	1.02 (0.58–1.78)		2.13 (1.27–3.57)	
**Sexual orientation**				
LGBTQA+	1.57 (0.72–3.39)	0.254	1.53 (0.74–3.14)	0.250
**Education**				
Secondary school or high-school diploma	1.305 (0.697–2.441)	0.405	1.099 (0.621–1.948)	0.745
**Professional status**				
Employed	Ref.		Ref.	
Student	1.60 (0.73–3.50)	0.242	2.29 (1.13–4.68)	**0.022**
Unemployed, retired, homemaker	0.49 (0.13–1.88)	0.298	0.87 (0.30–2.57)	0.809
**Nationality**				
Born abroad or parents born abroad (1 or 2)	0.49 (0.13–1.85)	0.294	1.00 (0.36–2.78)	0.995
**Family relationship**				
Conflictual/very conflictual	1.08 (0.58–2.00)	0.807	1.21 (0.36–2.78)	0.496
**Social life**				
Many people	Ref.		Ref.	
Some people	1.76 (0.93–3.32)	0.080	1.34 (0.79–2.27)	0.274
No contact	7.21 (1.94–26.80)	**0.003**	10.56 (2.64–42.26)	**0.001**
Social isolation	8.76 (1.99–38.44)	**0.004**	2.52 (0.63–9.99)	0.189
**Economic status**				
Low/very low	2.16 (1.00–4.65)	0.050	0.84 (0.39–1.83)	0.661
**CAGE-AID**				
Positive	2.14 (1.20–3.80)	0.385	1.72 (1.02–2.92)	**0.043**
**Being victim of CyA**	0.77 (0.44–1.38)	0.385	1.14 (0.69–1.88)	0.620
**Being perpetrator of CyA**	1.40 (0.64–3.11)	0.397	1.04 (0.50–2.18)	0.916
**Being victim of FtF-A ^2^**	0.89 (0.52–1.55)	0.692	1.19 (0.73–1.88)	0.476
**Being perpetrator of FtF-A ^2^**	0.57 (0.25–1.32)	0.188	0.90 (0.43–1.88)	0.778
**Access to mental health service**	1.91 (1.03–3.55)	**0.040**	2.42 (1.36–4.31)	**0.003**

Abbreviations: adjOR, adjusted odds ratio; CI, confidence interval; CyA, cyberaggression; FtF-A, face-to-face aggression; CAGE-AID, Cut Down, Annoyed, Guilty, and Eye Opener—Adapted to Include Drug Use; LGBTQA, lesbian, gay, bisexual, transgender, queer, asexual. Notes: ^1^ Female-to-male ratio and queer were omitted due to small size. ^2^ Dichotomous variable (never vs. during childhood, adulthood, or both). Significant *p*-values in bold.

## Data Availability

Raw data were generated at the Department of Public Health Sciences and Pediatrics, University of Turin. Derived data supporting the findings of this study are available from the corresponding author (M.M.) on request.

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
