# Peer review of "Exploring Cyberaggression and Mental Health Consequences among Adults: An Italian Nationwide Cross-Sectional Study"

_ijerph, 2023, doi:10.3390/ijerph20043224_

Round 1
Reviewer 1 Report
line 63 - is it "10 to 20%"?
line 112 - inconsistent citation format
146 - change was to were. Remove extra '('
306 - consider changing to "Methods section"
309 - please explain elimination of A+ from LBGTQA+ group
371 - is it 'concealed' or cancelled as in deleted profile?
323 & 376 - should it be "LBGTQ+"; also consider using word 'population' instead of 'people'
Author Response
"Please see the attachment."

Reviewer 2 Report
This study explored the potential consequences of Cyber-aggression on mental health of Italian adults, which was of great significance to the Internet era with frequent Cyber-aggression. Here are some suggestions:
Background: 1. The logic of the context (especially the connectives) needs to be considered again. 2. What do lines 36-41 want to express about the detailed description of Budapest Convention? 3. It is suggested to write the general background of the study first, then describe the definition of the research variables and the research status, and finally indicate the purpose of this study.
Discussion: 1. Is line 294 consistent with all previous studies? If not, it is recommended to add references. 2. Lines 297-305: It is recommended to describe the results of other studies and compare them with the results of this study. 3. The part of discussion should focus on discussing the new findings of this study and the similarities and differences with previous research results.
Reviewer 3 Report
The study adds to the literature by providing evidence of impacts of cyberbullying and cyberaggression in an adult sample in Italy. Studies have primarily focused on youth. A strength of the study is the asking by by-standing behavior in online bullying and aggression. The use of the English language is awkward in places and should be edited to improve word usage and grammar.
Maybe “perpetrator” is a better word to use for someone who initiate cyberbullying or cyberaggression than “author”. Perpetrator is used later in the manuscript so it would be good to be consistent in the earlier sections of the paper.
Was data available on race or ethnicity since this might be a source of discrimination leading to cyberaggression? Is this the purpose of place of birth of the respondent and their parents?
Was age not normally-distributed? It might be interesting to ask whether the mean or median age differed in perpetrators and victims of cyberaggression.
Could respondents report being both a victim and a perpetrator? It would be good to state the percentage that were victims only, perpetrators only, or both around line 199. How does this influence the statistical models when there is overlap in these categories? Were those endorsing both excluded from the analyses? Were there bystanders who also reported being a victim or perpetrator?
Where did the variables discussed in lines 244-249 come from? Many of them were not mentioned in the methods section.
In terms of being both a perpetrator and a victim, it might be that adults are more lkely to respond to a first strike aggressive attack online by attacking. People are less inhibited online and adults have no consequences to responding similarly. Is that what is occurring here? What might be the most revealing information is who initiated the first aggressive attack.
Can you provide an explanation for the differences in access to mental healthcare between victims and perpetrators? Could this be the way the question was worded. Was it asked such that it reflects using mental health services?
The discussion is clearly written. It might be good to highlight what is new in the current study and move it further up in the discussion.
Overall, organizations improvements could be made to make it flow better. For example, having a section header of “Chi-square analysis” is not very informative. Section titles could better describe what research question is being asked and answered.
Reviewer 4 Report
The detailed manuscript review comments are attached below.

Reviewer 5 Report
1. We found that the authors have published similar paper” Consequences of cyberaggression on Social Network on mental health of Italian adults” in Eur. J. Public Health, DOI : 10.1093/eurpub/ckab165.589, please describe the differences.
2. We found that women are more than men, did the difference between gender affect the results and conclusions?
3. As a nationwide cross-sectional study, could you please describe the distribution of different regions in the article? Did the difference between different regions affect the results of the study?
4. The sample size of the study was only 446, which might be the limitation.
5. What is the difference and association between CyA and CyV? Could you briefly explain them in the article?
6. A large part of the article is talking about CyV, what is the major content, CyA or CyV?
Reviewer 6 Report
Thank you for giving me the opportunity to review this interesting study. I have only a few minor comments, as described below:
1. How the sample size was calculated?
2. Where are the validity and reliability tests of the developed questionnaire?
3. Where are the implications and future recommendations for research?
Round 2
Reviewer 2 Report
The details of the article and the logic of the context have been greatly improved, and the discussion part has also been integrated with thinking and comparison.
Reviewer 3 Report
The manuscript has improved and is easier to read and understand. There are still places where the English needs improving. The remainder of my comments have been adequately addressed.
Reviewer 6 Report
The authors have addressed all of my comments